# Femoral Anteversion Measured by the Surgical Transepicondylar Axis Is Correlated with the Tibial Tubercle–Roman Arch Distance in Patients with Lateral Patellar Dislocation

**DOI:** 10.3390/medicina59020382

**Published:** 2023-02-16

**Authors:** Jiaxing Chen, Fuling Chen, Lin Fan, Sizhu Liu, Yi Feng, Qiaochu Li, Jian Zhang, Zhengxue Quan, Aiguo Zhou

**Affiliations:** 1Department of Orthopedics, The First Affiliated Hospital of Chongqing Medical University, Chongqing 400016, China; 2Orthopedic Laboratory, Chongqing Medical University, Chongqing 400016, China; 3Department of Orthopedics, People’s Hospital of Hechuan, Chongqing 401520, China; 4Department of Radiology, The First Affiliated Hospital of Chongqing Medical University, Chongqing 400016, China; 5Medical Education Department, The First Affiliated Hospital of Chongqing Medical University, Chongqing 400016, China

**Keywords:** TT-RA distance, femoral anteversion, surgical transepicondylar axis, patellar dislocation, tibial tubercle osteotomy

## Abstract

*Background and Objectives:* Various predisposing factors for lateral patellar dislocation (LPD) have been identified, but the relation between femoral rotational deformity and the tibial tubercle–Roman arch (TT-RA) distance remains elusive. *Materials and Methods:* We conducted this study including 72 consecutive patients with unilateral LPD. Femoral anteversion was measured by the surgical transepicondylar axis (S-tAV), and the posterior condylar reference line (P-tAV), TT-RA distance, trochlear dysplasia, knee joint rotation, patellar height, and hip–knee–ankle angle were measured by CT images or by radiographs. The correlations among these parameters were analyzed, and the parameters were compared between patients with and without a pathological TT-RA distance. Binary regression analysis was performed, and receiver operating characteristic curves were obtained. *Results:* The TT-RA distance was correlated with S-tAV (r = 0.360, *p* = 0.002), but the correlation between P-tAV and the TT-RA distance was not significant. S-tAV had an AUC of 0.711 for predicting a pathological TT-RA, with a value of >18.6° indicating 54.8% sensitivity and 82.9% specificity. S-tAV revealed an OR of 1.13 (95% CI [1.04, 1.22], *p* = 0.003) with regard to the pathological TT-RA distance by an adjusted regression model. *Conclusions:* S-tAV was significantly correlated with the TT-RA distance, with a correlation coefficient of 0.360, and was identified as an independent risk factor for a pathological TT-RA distance. However, the TT-RA distance was found to be independent of P-tAV.

## 1. Introduction

As a common sports-related disorder, lateral patellar dislocation (LPD) has a morbidity of about 108 per 100,000 individuals [1]. A variety of risk factors are involved in LPD, such as excessive tibial tubercle lateralization, valgus malalignment, and trochlear dysplasia [2,3]. Notably, femoral rotational malformation is common in patients with LPD and is reported as the primary bony factor producing patellofemoral instability [4,5]. When femoral malrotation is demonstrated as the main etiology of LPD, derotational distal femoral osteotomy (DDFO) is considered an appropriate operation [6,7].

Accurately evaluating the anatomic abnormalities of patients with LPD is of great significance. In 2022, a systematic review highlighted the importance of an accurate radiological evaluation to perform an individualized surgical strategy [8]. Using the posterior condylar reference line (P-tAV) to measure femoral anteversion cannot allow an accurate evaluation of the rotational malformation due to the influence of posterior femoral condylar dysplasia; the surgical transepicondylar axis (SEA) serves as a more appropriate reference line for evaluating femoral anteversion (S-tAV) [9]. The morphologies of the posterior femoral condyles rarely influence the measurement of S-tAV, with a value of >20.4° indicating pathology. In addition, the tibial tubercle–trochlear groove (TT-TG) distance is significantly influenced by the trochlear morphology when evaluating tibial tubercle lateralization, and the tibial tubercle–Roman arch (TT-RA) distance was reported as a more reliable parameter, with a pathology corresponding to a value of >26 mm [10].

A controversy exists in the literature with regard to the relation between femoral anteversion and the lateralization of the tibial tubercle. Wang et al. [11] found that the TT-TG distance was positively correlated with P-tAV. However, Xu et al. [4] and Prakash et al. [12] reported that the TT-TG distance was not significantly associated with P-tAV. In addition, femoral anteversion has been demonstrated to have a positive effect on the measurement of the TT-TG distance [13]. Improper parameter selection for evaluating femoral anteversion or tibial tubercle lateralization might be a reason for such discrepancy. However, the relationship between the TT-RA distance and S-tAV has not been reported yet.

Given that the relation between femoral rotational deformity and the parameters for evaluating tibial tubercle lateralization remains controversial in the literature, this study aimed to verify the relationship between the TT-RA distance and femoral anteversion in patients with LPD. We hypothesized that the TT-RA distance could be directly correlated with S-tAV.

## 2. Materials and Methods

### 2.1. Study Population

The ethics committee of our hospital approved this study and waived the informed consent (IRB number: 2022-K8; Approval date: 5 January 2022). In total, 135 patients were initially identified from January 2016 to December 2021 in our institution from the Electronic Medical Records System based on the inclusion criterion of unilateral and recurrent patellar dislocators with skeletal maturity. Patients were excluded from this study if they were without sufficient or standard radiological data, had hip developmental dysplasia, previously suffered from trauma or surgery, or presented with distal femoral condyle epiphysitis, which could influence the reliability of the measurements.

In total, 36 patients did not have CT scans of the hip, femur, and knee joint simultaneously; 13 patients had nonstandard weight-bearing full-leg anteroposterior radiographs; 3 patients suffered from trauma or surgery; 2 patients presented severe condyle epiphysitis; 4 patients showed hip developmental dysplasia. Finally, 77 patients were enrolled in this study for radiological assessment.

### 2.2. Computed Tomography Technique

A CT scanner (Somatom Sensation, Siemens Healthcare, Forchheim, Germany) was used to obtain images of the patients ranging from the ilium to the toes. All the patients were supine on the examination bed during the CT scan, the knee joints were fully extended, and the ankle joints were in a neutral position. Important scanning parameters: according to the size of the extremities of the patients, the field of view varied from 220 to 450 mm; both the scanning layer thickness and the layer spacing were 1 mm, with a matrix of 512 × 512 pixels.

### 2.3. Radiological Assessment

We included 77 patients with LPD for radiological assessment. All the measurements were retrospectively conducted by a radiologist and an orthopedist on CT images and radiographs via a picture archiving and communication system (PACS). The observers received professional training regarding the measuring method of each parameter. Major disputes of the measuring results, especially on the Dejour classification of trochlear dysplasia, were discussed until consensus was reached. In this study, the anatomic parameters were retrospectively evaluated by PACS, and the accuracy of each measurement was one digit behind a comma.

### 2.4. Measurements

#### 2.4.1. Femoral Anteversion

According to the method described previously, the segmental torsion parameters of the femur were measured by CT images [14]. Three different transverse slices of the femur were identified to draw four reference lines: a line a passing through the center of the femoral head and neck, a distal femoral shaft line, a posterior condylar reference line (PCRL), and the SEA (Figure 1). P-tAV (formed between line a and the PCRL) and S-tAV (formed between line a and the SEA) represent the total malrotation of the femur. The angle between the distal femoral shaft line and the PCRL (P-dAV) or the SEA (S-dAV) reflects the rotational malformation of the distal femur.

#### 2.4.2. Trochlear Dysplasia

Trochlear dysplasia includes four subtypes identified by axial CT slice according to the Dejour classification [15]. Type A (mild) shows a relatively shallow trochlea; type B shows a flat or convex trochlea; type C has a convex medial trochlear wall and asymmetric facets; type D (severe) presents asymmetric trochlear facets with a cliff. The LTI was a reliable parameter for objectively evaluating trochlear dysplasia, with a value of less than 11° indicating trochlear dysplasia [16]. It is the angle formed between the PCRL and the tangent line of the lateral trochlear facet (Figure 2).

#### 2.4.3. TT-RA Distance

Referring to the method that we described previously, axial CT images were used to measure the TT-RA distance [10]. In short, the axial CT slice showing an intact Roman arch and the posterior femoral condyles was selected to mark the highest point of the Roman arch. The axial slice at the insertion of the patellar tendon was identified to determine the midpoint of the tibial tubercle. With the PCRL as the reference line, the distance between the highest point of the Roman arch and the midpoint of the tibial tubercle was measured to define the TT-RA distance (Figure 3).

#### 2.4.4. Knee Joint Rotation and Patellar Height

Knee joint rotation (KJR) is defined as the angle formed between the PCRL and the proximal tibial condyle reference line in the slice with visualization of the intact posterior tibial condyles [17] (Appendix A). The Insall–Salvati index (ISI) can reliably evaluate the patellar height on standard lateral radiographs and is calculated as the patellar tendon divided by the maximum length of the patella measured from the distal pole to the proximal pole, with a value greater than 1.2 indicating patella alta [18] (Appendix A).

#### 2.4.5. Coronal Malalignment

Coronal malalignment was assessed by hip–knee–ankle (HKA) angle on weight-bearing full-leg radiographs. It is defined as the angle between the femoral mechanical axis and the tibial mechanical axis [19] (Figure 4). Valgus deformity was diagnosed with a value of the HKA angle >1.5°.

### 2.5. Statistical Analysis

We calculated the interclass correlation coefficient (ICC) to evaluate the inter-observer reliability of each measurement; a value higher than 0.75 indicates an excellent agreement. To exclude abnormal values from the final analysis, box plot analysis was conducted. The average value of each parameter was utilized, and all statistical analyses were performed by an experienced orthopedist via the SPSS software (version 21.0; IBM Corp, Armonk, NY, USA). We performed the Shapiro–Wilk normality test to verify if the data conformed to the normal distribution. Spearman or Pearson correlation analysis was conducted to deal with the relation between the anatomic parameters. Chi-square test, Wilcoxon rank-sum test (for data not conforming to the normal distribution), and unpaired t-test (for data conforming to the normal distribution) were used to identify the differences in the anatomic parameters between the two subgroups. GraphPad Software (version 8.0.2, San Diego, CA, USA) was used to conduct receiver operating characteristic curves (ROC) analysis. The areas under the curves (AUC) and the Youden index were calculated to determine the ability of the parameters to predict a pathological TT-RA distance. We established a binary logistic regression model to identify the relationship between femoral anteversion and the TT-RA distance. *p* < 0.05 represents statistical significance.

The power of this study was calculated by G-Power software (version 3.1.9.4, Dusseldorf, Germany). For the effective size of 0.36 calculated by the correlation coefficient between the TT-RA distance and S-tAV in this study, the sample size was 72, and the alpha value was set to 0.05; this study obtained a power of 0.94.

## 3. Results

Five patients with abnormal data were excluded from the final statistical analysis by box plot. A total of 72 subjects were eventually enrolled in the statistical analysis; they included 57 females and 15 males (mean age ± SD 21.9 ± 8.4). The detailed demographic data are shown in Table 1. The ICC values of each measurement for inter-observer reliability showed an excellent agreement (Table 2). Except for the HKA angle, all the anatomic parameters were consistent with the normal distribution.

S-tAV was correlated with S-dAV in patients with LPD (r = 0.290, *p* = 0.013), similar to what observed for the relation between P-tAV and P-dAV (r = 0.333, *p* = 0.004). S-tAV was associated with the TT-RA distance (r = 0.360, *p* = 0.002) and the HKA angle (r = 0.319, *p* = 0.006). P-tAV was correlated with patellar height (r = 0.242, *p* = 0.040) and HKA angle (r = 0.270, *p* = 0.022). The correlation between P-dAV and the HKA angle showed statistical significance (r = 0.255, *p* = 0.031). Table 3 shows the results of the correlation analysis for the anatomic parameters.

According to the TT-RA distance, the patients with a value higher than 26 mm and smaller than 26 mm were designed as group A and group B, respectively. The results of a different analysis of the parameters between the two subgroups are shown in Table 4. S-tAV was larger in group A (19.0 ± 7.8 degrees) than in group B (13.1 ± 6.8 degrees) (*p* = 0.001). The difference in P-tAV between the two subgroups showed no statistical significance. There were no significant differences in the rest of the parameters between the two subgroups (*p* > 0.05).

As shown by the results of the ROC curve analysis, S-tAV had a significant AUC of 0.711 for predicting a pathological TT-RA distance (*p* = 0.002), with a value of >18.6° indicating 54.8% sensitivity and 82.9% specificity. P-tAV had an AUC of 0.644 for a pathological TT-RA distance (*p* = 0.038) (Figure 5). S-tAV had an OR of 1.12 (95% CI [1.04, 1.20], *p* = 0.002) for predicting a pathological TT-RA distance by simple binary regression analysis. A multivariable analysis (the HKA angle and S-dAV were included) showed that S-tAV was associated with the TT-RA distance, with an OR of 1.13 (95% CI [1.04, 1.22], *p* = 0.003) (Figure 6).

## 4. Discussion

The most significant findings of this study are here described. The TT-RA distance was directly correlated with S-tAV but was independent of P-tAV in patients with LPD. S-tAV had a fair diagnostic ability for predicting a pathological TT-RA distance and was also verified as an independent risk factor for a pathological TT-RA distance. When the value of S-tAV was greater than 18.6°, the sensitivity to identify a pathological TT-RA distance was 54.8%. As S- tAV increased by one degree, the risk of a pathological TT-RA distance was 1.13-fold higher. These findings indicate the significant correlation between the lateralization of the tibial tubercle and femoral malrotation.

Femoral rotational malformation was reported as a significant etiology for LPD by decreasing the patellofemoral congruence and increasing the force on the lateral patellofemoral facet [20,21]. If femoral rotational malformation is left uncorrected, the tension of the reconstructed medial patellofemoral ligament would increase, followed by patellar re-dislocation [22]. In addition, excessive femoral anteversion could occur with other skeletal abnormalities, such as excessive lateralization of the tibial tubercle [23]. The TT-TG distance has been widely used to evaluate the tibial tubercle lateralization, and the relationship between the TT-TG distance and femoral rotational deformities was investigated by some researchers, but different results were reported [11].

One of the reasons for such discrepancies might be that the parameters selected to evaluate a specific deformity could not accurately reflect the studies malformation. The measurement of the TT-TG distance is significantly dependent on trochlear morphology, because the lowest point of the trochlea is hard to identify, especially in patients with severe trochlear dysplasia. Because of the impact of the morphology of the posterior femoral condyles, P-tAV could potentially over-evaluate a femoral rotational deformity [24,25]. Then, the TT-RA distance and S-tAV were proposed by researchers as alternatives of the TT-TG distance and P-tAV to accurately evaluate the tibial tubercle lateralization and femoral malrotation, respectively [10]. On the other hand, few researchers examined with the relation between the lateralization of the tibial tubercle and femoral anteversion. Thus, the correlation between S-tAV and the TT-RA distance warrants urgent investigation to address such gap in the literature.

Prakash et al. [12] found that P-tAV showed no correlation with the TT-TG distance (r = −0.09). Havey et al. [26] demonstrated that the TT-TG distance was not associated with femoral anteversion (r = 0.103) in patients with patellofemoral instability. Hochreiter et al. [27] used the transepicondylar axis to measure the femoral anteversion in patients with knee osteoarthritis and indicated that the TT-TG distance was not associated with femoral anteversion. Diederichs et al. [21] found that the correlation between the segmental femoral rotational parameters and the TT-TG distance showed no significance. Dickschas et al. [13] revealed a non-significant correlation between femoral anteversion and the TT-TG distance but proposed that the measurement of the TT-TG distance could be influenced by femoral anteversion. In contrast, Wang et al. [11] dealt with the correlation between P-tAV and the TT-TG distance and did report a significant correlation. In this study, the correlation between P-tAV and the TT-RA distance was not statistically significant, which is in line with our previous results [4]. Notably, the TT-RA distance was significantly correlated with S-tAV, with an r value of 0.36, indicating that the femoral anteversion angle has a positive effect on the measurement of the lateralization of the tibial tubercle, the TT-RA distance. To the best of our knowledge, the specific correlation between the TT-RA distance and femoral anteversion was first identified in this study: when the value of S-tAV was greater than 18.6°, the sensitivity of detecting a pathological TT-RA distance was 54.8%, and as S- tAV increased by one degree, the risk of a pathological TT-RA distance was 1.13-fold higher.

The literature regarding the relation between the TT-TG distance and distal femoral anteversion is scarce. Distal femoral anteversion was considered a contributor to excessive total femoral malrotation in patients with LPD [14]. In this study, the rotational deformity of the distal femur was positively correlated with total femoral anteversion. Previously, we found that the distal femoral anteversion measured by the PCRL (P-dAV) was significantly correlated with the TT-TG distance [4]. However, based on the data of our study with a larger sample size, both P-dAV and S-dAV showed no association with the TT-RA distance. The different parameters here considered with respect to the previous study might be a reason for such discrepancy; additionally, another possible reason is that distal femoral anteversion is not the only etiology producing total femoral malrotation [4], and the relationship of segmental femoral anteversion with the TT-RA distance warrants further investigation.

As it stands, femoral rotational abnormalities in patients with LPD could be alleviated via DDFO to some extent [28]. Nevertheless, Kaiser et al. [29] elucidated that DDFO would change the recorded value of the TT-TG distance postoperatively and suggested that when considering surgical procedures, the relation between the TT-TG distance and femoral anteversion should not be neglected. The preoperative correlation between the TT-RA distance and femoral anteversion was specified in our study, but the variation tendency of the TT-RA distance after DDFO warrants further investigation, which could help orthopedic surgeons to choose an appropriate surgical strategy. For patients with LPD in the presence of excessive femoral anteversion and increased TT-RA distance preoperatively, if the value of the TT-RA distance decreases to a normal range after DDFO, the necessity of additional tibial tubercle osteotomy is worthy of studying.

Trochlear dysplasia has been accepted as the most significant predisposing factor for LPD, and the Dejour classification and LTI are commonly used to evaluate trochlear dysplasia [15]. P-dAV showed a significant correlation with Dejour classification [11]. Previous research showed that P-tAV was correlated with LTI, and distal femoral anteversion showed a significant correlation with Dejour classification in patients with LPD [30]. Diederichs et al. [21] found that segmental femoral anteversion, such as P-tAV and P-dAV, was not associated with trochlear dysplasia. Based on our data, the relationship between femoral anteversion and trochlear dysplasia was not statistically significant.

Knee valgus deformity contributes to patellar instability by increasing the Q angle and lateral vector on the patella, and the HKA angle is an accepted parameter for evaluating the coronal malalignment of the lower extremities [31]. Li et al. [32] found that P-tAV showed a strong correlation with the HKA angle. Zhang et al. [20] reported that the valgus angle did not vary from patients with normal P-tAV to patients with pathological P-tAV (>30°). In our study, the HKA angle was correlated with both P-tAV and S-tAV. However, the specific etiology of valgus deformity remains elusive in patients with LPD. Eberbach et al. [33] indicated that valgus deformity was attributable to a tibial malformation in the majority of patients. In addition, Gillespie et al. [31] reported that the morphologies of distal femur and proximal tibia were correlated with lower limb alignment in patients with LPD. The abnormalities (femoral-based, tibial-based, or intra-articular-based abnormalities) contributing to knee valgus deformity need further investigation.

Patella alta was also considered an important risk factor for LPD [15]. Research regarding the relation between patellar height and femoral anteversion is scarce. Nelitz et al. [34] conducted a study including 12 patients with severe femoral rotational deformity and reported that the median value of the ISI and P-tAV were 1.2 and 33.8°, respectively. Diederichs et al. [21] found that the ISI was not correlated with femoral anteversion. Based on our data, the ISI was poorly correlated with P-tAV (r = 0.242). However, whether other parameters for evaluating patellar height, such as the Caton–Deschamps index, could obtain the same results warrants further studying.

Our study has some limitations. First, not all segmental femoral torsion parameters were analyzed, such as neck and mid femoral anteversion. Second, the tibial morphology was not included, which has been demonstrated to be associated with LPD [35]. Third, measurements by MRI were not performed. Fourth, this is a retrospective case series study and did not enroll healthy controls; the results calculated on the basis of the data of our study may change is a larger sample or patients of different races are included. Fifth, except for the LTI, the quantitative parameters for evaluating trochlear dysplasia were not studied, such as trochlear depth and sulcus angle. Sixth, the anatomical features of the femoral anteversion and the TT-RA distance in patients with growth potential are unclear. Seventh, intra-observer agreements of the parameters were not determined, while we assumed the intra-observer bias to be minimal because both the orthopedist and the radiologist were well trained on the measuring methods, and the parameters included in this study were found to have excellent reliability values by recent studies [10,15].

## 5. Conclusions

S-tAV was significantly correlated with the TT-RA distance, with a correlation coefficient of 0.360, and appeared as an independent risk factor for a pathological TT-RA distance. However, P-tAV was shown to be independent of the TT-RA distance.

## Figures and Tables

**Figure 1 medicina-59-00382-f001:**
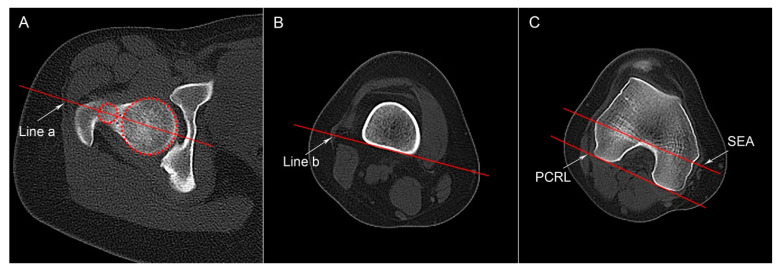
Segmental femoral torsion parameters. (**A**) An axial CT slice showing the intact femoral head and neck is selected. The two circles were drawn to the margin of femoral head and neck to identify the center of the head and neck; Line a passes through the center of both the femoral head and the neck. (**B**) Distal femoral shaft line (Line b) is tangent to the posterior bony margin of the femoral shaft on the slice above the gastrocnemius insertion. (**C**) An axial slice showing an intact “Roman Arch” and the femoral condyles. The posterior condylar reference line (PCRL) was drawn tangent to the posterior femoral condyles; the surgical transepicondylar axis (SEA) passes through the sulcus of the medial epicondyle and the prominence of the lateral epicondyle.

**Figure 2 medicina-59-00382-f002:**
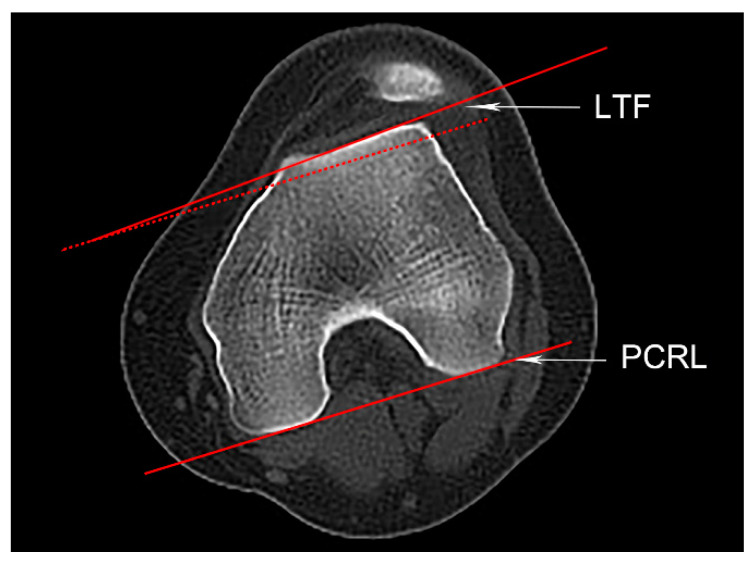
Lateral trochlea inclination (LTI). An axial slice showing an intact “Roman Arch” and the femoral condyles. The posterior condylar reference line (PCRL), its parallel line (the red dotted line), and a tangent line of the lateral trochlear facet (LTF) are shown. The LTI is defined as the angle between the PCRL and the LTF.

**Figure 3 medicina-59-00382-f003:**
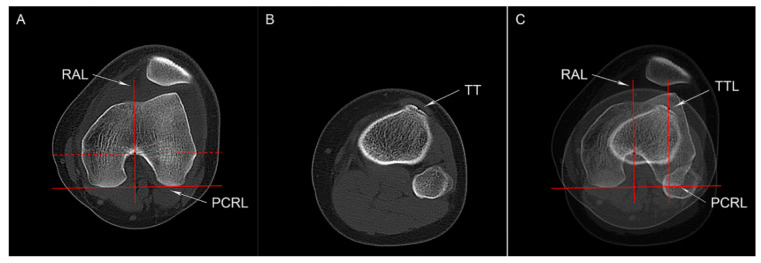
Tibial tubercle to Roman arch (TT-RA) distance. (**A**) The PCRL was drawn tangent to the posterior femoral condyles, and its parallel line (the doted one) was tangent to the Roman arch; a line perpendicular to the PCRL was drawn to pass through the tangent point (RAL). (**B**) The center of the tibial tuberosity (TT) is marked on an axial CT slice at the insertion of the patellar tendon. (**C**) After superimposing the two images, the line parallel to the RAL was drown through the TT (TTL). The TT-RA distance is defined as the distance between the RAL and the TTL.

**Figure 4 medicina-59-00382-f004:**
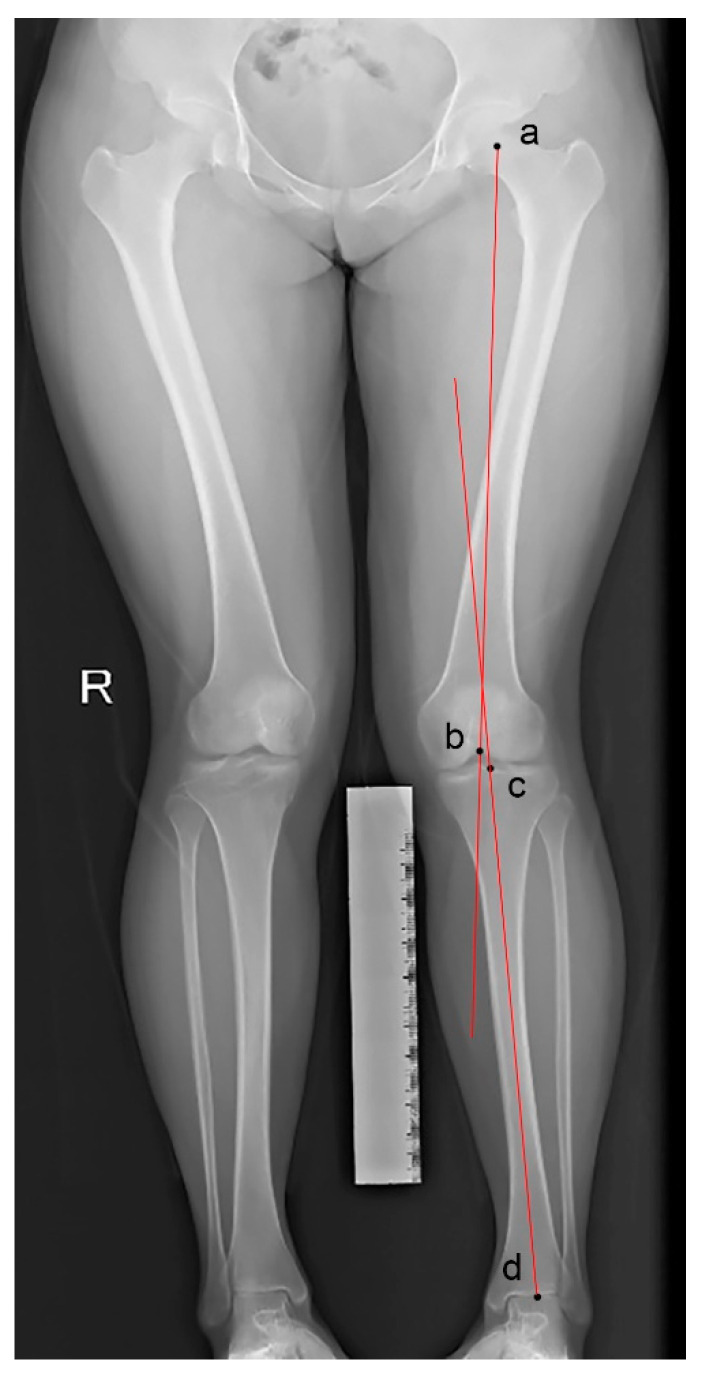
Hip–knee–ankle (HKA) angle measured by a weight-bearing full-leg radiograph. The points represent the center of the femoral head (a), the femoral condyles (b), the tibial plateau (c), and the ankle joint (d). It is the angle formed between the femoral axis (ab) and the tibial axis (cd).

**Figure 5 medicina-59-00382-f005:**
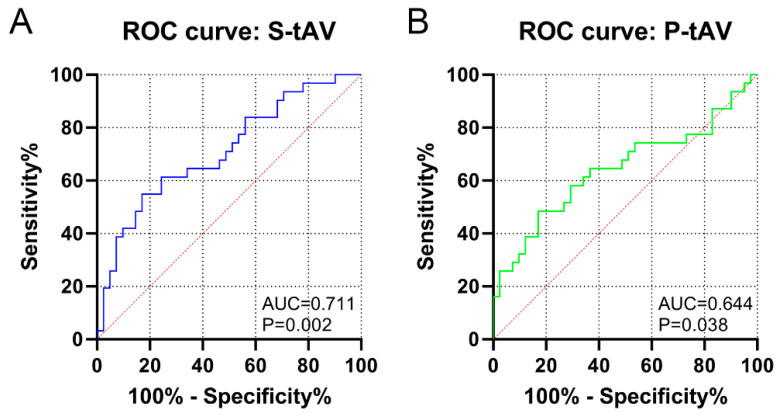
Receiver operating characteristic curve (ROC) of S-tAV (**A**) and P-tAV (**B**) for a pathological TT-RA distance. The areas under the curves (AUC) and the *p* values are shown.

**Figure 6 medicina-59-00382-f006:**
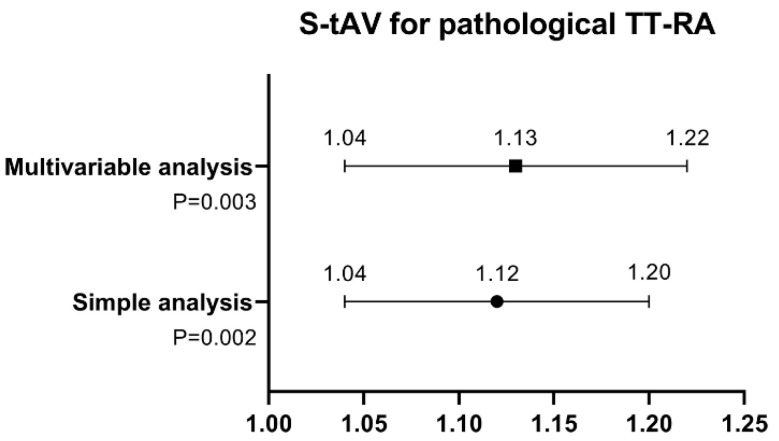
Binary regression model for a pathological tibial tubercle–Roman arch distance. The odds ratio, 95% confidence interval, and *p* values are shown.

**Table 1 medicina-59-00382-t001:** Demographic data of the included patients.

Variables	
Number of patients, n	72
Sex, (Female/Male), n	57/15
Age, mean ± SD, y	21.9 ± 8.4
Side of knee, n	
Left	35
Right	37
Dejour classification, n	
A/B/C/D	8/15/27/22

SD, standard deviation.

**Table 2 medicina-59-00382-t002:** ICC values of the anatomic parameters for inter-observer reliability, showing mean ± SD.

	Observer 1	Observer 2	ICC	95% CI
Femoral torsion				
S-tAV, °	15.4 ± 8.0	16.5 ± 7.3	0.985	0.976, 0.990
P-tAV, °	19.6 ± 7.5	20.2 ± 6.8	0.985	0.977, 0.991
S-dAV, °	10.4 ± 4.2	10.6 ± 3.9	0.939	0.906, 0.961
P-dAV, °	16.5 ± 3.9	17.0 ± 3.5	0.930	0.892, 0.955
TT-RA distance, mm	19.8 ± 3.7	20.1 ± 3.7	0.857	0.784, 0.907
Dejour classification, n ^a^ (A/B/C/D)	7/14/29/22	8/14/25/25	0.774	0.678, 0.871
LTI, °	10.3 ± 5.7	10.5 ± 5.1	0.945	0.914, 0.964
Knee joint rotation, °	8.0 ± 4.4	8.4 ± 4.3	0.957	0.933, 0.972
Patellar height	1.30 ± 0.17	1.28 ± 0.13	0.882	0.820, 0.923
HKA angle, °	1.8 ± 2.1	2.0 ± 2.0	0.971	0.955, 0.981

S-tAV and P-tAV, total femoral anteversion measured by the surgical transepicondylar axis (SEA) and the posterior condylar reference line (PCRL), respectively; S-dAV and P-dAV, distal femoral anteversion measured by the SEA and the PCRL, respectively; TT-RA distance, tibial tubercle to Roman arch distance; LTI, lateral trochlea inclination; HKA, hip–knee–ankle angle; SD, standard deviation; ICC, interclass correlation coefficient; CI, confidence interval; ^a^, showing the results of weighted kappa analysis.

**Table 3 medicina-59-00382-t003:** Relation between the femoral torsion parameters and other anatomic parameters.

	TT-RA Distance	Dejour Classification ^a^	LTI	Knee Joint Rotation	Patellar Height	HKA Angle ^a^
S-tAV	0.360 **	0.132	0.032	−0.060	0.215	0.319 **
P-tAV	0.229	0.164	−0.091	0.091	0.242 *	0.270 *
S-dAV	0.122	−0.123	0.022	−0.180	−0.105	0.125
P-dAV	0.051	−0.068	0.057	−0.241	0.004	0.255 *

S-tAV and P-tAV, total femoral anteversion measured by the surgical transepicondylar axis (SEA) and the posterior condylar reference line (PCRL), respectively; S-dAV and P-dAV, distal femoral anteversion measured by the SEA and the PCRL, respectively; TT-RA distance, tibial tubercle to Roman arch distance; LTI, lateral trochlea inclination; HKA, hip–knee–ankle angle; *, significance with a *p* value of <0.05; **, significance with a *p* value of <0.01; ^a^, showing the results of Spearman correlation analysis.

**Table 4 medicina-59-00382-t004:** Difference in the variables between the two examined subgroups, showing the mean and standard deviation.

	TT-RA ≥ 26 mm (n = 31)	TT-RA < 26 mm (n = 41)	*p* Value
Sex (female, n)	24	33	0.751 ^a^
Age, y	22.5 ± 9.4	21.5 ± 7.5	0.601
Femoral torsion			
S-tAV, °	19.0 ± 7.8	13.1 ± 6.8	0.001
P-tAV, °	21.2 ± 7.8	18.1 ± 6.1	0.055
S-dAV, °	11.0 ± 4.3	10.1 ± 3.3	0.339
P-dAV, °	16.9 ± 3.7	16.4 ±3.0	0.776
TT-RA distance, mm	28.0 ± 2.0	22.3 ± 1.8	<0.001
Dejour classification, n (A/B/C/D)	5/5/11/10	3/10/16/12	0.589 ^a^
LTI, °	10.0 ± 4.5	10.4 ± 5.0	0.789
Knee joint rotation, °	8.2 ± 4.0	8.3 ± 4.8	0.974
Patellar height	1.28 ± 0.14	1.30 ± 0.15	0.671
HKA angle, °			0.678 ^b^
Median	1.7	2.0	
IQR	2.1 (0.7–2.8)	1.9 (1.2–3.1)	

S-tAV and P-tAV, total femoral anteversion measured by the surgical transepicondylar axis (SEA) and the posterior condylar reference line (PCRL), respectively; S-dAV and P-dAV, distal femoral anteversion measured by the SEA and the PCRL, respectively; TT-RA distance, tibial tubercle to Roman arch distance; LTI, lateral trochlea inclination; HKA, hip–knee–ankle angle; IQR, interquartile range; ^a^, showing the results calculated by the Chi-square test; ^b^, showing the results calculated by the Wilcoxon rank-sum test.

## Data Availability

The data associated with the paper are not publicly available but are available from the corresponding author upon reasonable request.

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
