# Peer review of "Femoral Anteversion Measured by the Surgical Transepicondylar Axis Is Correlated with the Tibial Tubercle–Roman Arch Distance in Patients with Lateral Patellar Dislocation"

_medicina, 2023, doi:10.3390/medicina59020382_

Round 1
Reviewer 1 Report
line 79, 94 etc - the amount of patients is stressed. If so, can the different amount change the results. Has to be explained and proved. The significance of the results is more important than the number of patients.
line 86 SOMATOM has a spatial resolution 0.25 mm (https://www.siemens-healthineers.com/computed-tomography/dual-source-ct/somatom-force). This parameter has an influence on the results. The latter have to be validated against the spatial resolution.
table 2, etc : measurement units for torsion are not identified; ICC has three digits behind the coma. This has to be motivated.
line 343 The conclusion must provide correlation indexes and their significance.
Reviewer 2 Report
Congratulation to the authors.
The study is well structured, presents a clear analysis of the correlation between different parameters used to study femoral antiversion and the TT-RA distance.
The TT-RA distance is a more recent and, according to some authors, a more reliable parameter than the TT-TG distance for evaluating tibial tubercle lateralization being TT-RA distance independent of the femoral trochlear morphology. Data collection and statistical analysis are presented in detail, and the sample size is appropriate for the study.The results are presented clearly and adequately.
The discussion and the conclusions are congruous with the results obtained.
The limits of the study are well stated and this helps to take into due consideration the results obtained.
The correlation between The Femoral Anteversion Measured by Surgical Transepicondylar Axis and the Tibial Tubercle-Roman Arch Distance is interesting and seems worth consideringin patients with lateral patellar dislocation.
Reviewer 3 Report
The importance and soundness of the proposed hypotheses is really interested. Radiological analysis is well done and strictly analyzed. I'd like to read more about the utility of your work in clinical and surgical practice. Statistical analysis is very clear.
